# Research

materials science/nanotechnology

photoresist, molecular glass photoresist, electron beam lithography, photolithography, negative-tone photoresist

**Authors for correspondence:**
Shuangqing Wang
e-mail: g1704@iccas.ac.cn
Guoqiang Yang
e-mail: gqyang@iccas.ac.cn

This article has been edited by the Royal Society of Chemistry, including the commissioning, peer review process and editorial aspects up to the point of acceptance.

# Negative-tone molecular glass photoresist for high-resolution electron beam lithography

Yafei Wang, Long Chen, Jiating Yu, Xudong Guo, Shuangqing Wang and Guoqiang Yang

Beijing National Laboratory for Molecular Sciences, Key Laboratory of Photochemistry, Institute of Chemistry, University of Chinese Academy of Sciences, Chinese Academy of Sciences, Beijing 100190, People's Republic of China

SW, 0000-0002-8281-9399; GY, 0000-0003-0726-2217

A low molecular weight organic compound containing bisphenol A backbone (BPA-6OH) is reported as a negative-tone photoresist. This material has a high glass transition temperature and excellent thermal stability. A good contrast, well-resolved line pattern around 73.4 nm and sensitivity of 52 μC cm$^{-2}$ can be received for negative-tone molecular glass photoresist upon exposure in electron beam lithography system. It indicates that the negative-tone molecular glass photoresist is one of the promising candidates for use in electron beam lithography.

## 1. Introduction

Modern high-performance electronic devices require integrated circuits to be smaller and smaller, more efficient, and more powerful. This is commonly referred to as Moore's Law, which is an empirical rule that states that the number of transistors on a silicon chip doubles every 18–24 months [1,2]. Advanced photolithography technology leads to higher integration and shrinking size [3–5]. Photolithography has evolved from G-line (436 nm), I-line (365 nm) lithography, to deep ultraviolet (248 nm, 193 nm), and then to more advanced lithography–extreme ultraviolet lithography [6–8], and electron beam lithography (EBL) is also a competitive option for high integration. The critical dimension (CD) of the integrated circuit ranges from micrometre, submicrometre to nanometre. At the same time, the corresponding photoresists have also been studied and developed [9–16].

As the CD continues to shrink, roughness, pattern collapse and acid diffusion become extremely non-negligible compared

(a)                                                                    (b)

**Figure 1.** Chemical structures of the negative-tone molecular glass photoresist ((a) BPA-6OH; (b) TMMGU).

with conventional polymeric photoresists. Researchers have developed a series of novel photoresists, of which molecular glass photoresist is one of the promising candidates for advanced lithography. Compared with the traditional polymer resists, the molecular glass photoresists have attracted extensive attention and have an explicit molecular structure, low molecular weight and mono-dispersity. Moreover, the molecular glass photoresists have the advantages of high thermal stability, high glass transition temperature and amorphous state, ensuring the formation of a uniform and dense amorphous film [17–22].

Electron beam lithography (EBL) is a fundamental technique of nanofabrication, which can directly manufacture nanostructures smaller than 10 nm and produce arbitrary patterns. In addition, EBL is a vital technology for manufacturing photomasks [23–26]. These characteristics distinguish electron beam lithography from a wide variety of other lithography technologies. The most common electron beam resists are non-chemically amplified resists, such as PMMA [27], HSQ [28] and ZEP [29]. They possess high resolution at the cost of the sensitivity. In the chemically amplified resist (CAR) system, the catalyst, such as the photoacid, is produced by the photoacid generator undergoing the photochemical reactions after irradiation, which can interact with the surrounding matrix to trigger a cascade of chain reactions. Consequently, the multiple resist exposure events can be caused by a single irradiation, thereby increasing the photosensitivity. Compared with non-CAR, the number of photons, or dose, required to achieve equivalent solubility change in CAR is greatly reduced [21]. For polymer-based resist, such as polymethylmethacrylate (PMMA), the researcher found that the sensitivity of polymer-based resist primarily depends on the nature of the polymer, molecular weight and molecular weight distribution [30], and it is also influenced by the lithographic processes. PMMA is well known as the industry standard high-resolution resist for e-beam lithography. The sensitivity of positive-tone PMMA was reported to be about 350 $\mu C\,cm^{-2}$ using 50 keV acceleration voltage [31]. In most cases, the sensitivity needs to be improved in the practical application for a high output, which may need much more exposure time. Electron beam photoresists with high sensitivity and high resolution need to be developed.

Chemically amplified photoresists have been widely used up to now, they use a photoacid generator to form a latent image upon exposure with high sensitivity [32,33]. Our group has reported a novel chemically amplified positive-tone photoresist based on bis-phenol A structure for extreme ultraviolet lithography [9]. Here, we report a negative-tone molecular glass photoresist based on bis-phenol A backbone (BPA-6OH) for electron beam lithography. The negative-tone molecular glass photoresist gives a resolution as small as 73.4 nm, sensitivity of 52 $\mu C\,cm^{-2}$, which may be one of the promising candidates for electron beam lithography. Figure 1 shows the chemical structures of the negative-tone molecular glass photoresist.

# 2. Experimental

## 2.1. Materials

The synthesis processes of BPA-6OH have been published elsewhere [34]. The details of the synthesis and characterization are described below.

Tetrabromobisphenol A and methyl iodide reacted to get dimethyl tetrabromobisphenol A, the corresponding dimethyl tetrabromobisphenol A reacted with 4-methoxybenzene boronic acid to get 2,2-di(4-methoxy-3,5-di(p-methoxy phenyl)phenyl)propane, the last synthesis procedures are the same as described in the reference [34]. Then, Boron tribromide (22.5 g, 90 mmol) and dichloromethane

(40 ml) were added to a 500 ml three-neck flask and cooled to 0°C in an ice-water bath. 2,2-Di(4-methoxy-3,5-di(p-methoxy phenyl)phenyl)propane (6.81 g, 10 mmol) was slowly added to the solution, and the reaction mixture was stirred for 12 h. The distilled water (150 ml) was added slowly with stirring. Then, ethyl acetate (100 ml) was added to the reaction system, the organic portions were extracted and dried with MgSO4, filtered and rotary evaporated. The concentrated organic solution was poured into $n$-hexane (300 ml), stirred for 2 h and filtered to obtain the product. The product was dried to get yellow power BPA-6OH (5.05 g, yield: 84.6%). $^1$H-NMR (400 MHz, acetone-d$_6$) $\delta$ 8.37 (s, 4 H), 7.36 (d, $J = 8.5$ Hz, 8 H), 7.14 (s, 4 H), 6.86 (d, $J = 8.5$ Hz, 8 H), 6.72 (s, 2 H), 1.75 (s, 6 H). MS (MALDI-TOF) $m/z$ calcd. for ($C_{39}H_{32}O_6$): 596.219340. Found: 596.219676 (see electronic supplementary material, figures S1 and S2).

Triphenylsulfonium perfluoro-1-butanesulfonate (TPS-PFBS) was purchased from Merck Life Science (Shanghai). 1,3,4,6-tetrakis(methoxymethyl)glycoluril (TMMGU) was obtained from TCI (Shanghai). Propylene glycol monomethyl ether acetate (PGMEA) and tetramethylammonium hydroxide (TMAH) were purchased from Beijing Kempur Microelectronics Inc. All chemicals were used without further purification.

## 2.2. Lithography evaluation

BPA-6OH, triphenylsulfonium perfluoro-1-butanesulfonate (TPS-PFBS) (7.5 wt% relative to BPA-6OH), and cross-linker, 1,3,4,6-tetrakis(methoxymethyl)glycoluril (TMMGU) (20 wt% relative to BPA-6OH) were dissolved in propylene glycol monomethyl ether acetate (PGMEA) to form a 40 mg ml$^{-1}$ solution. The solution was filtered three times through 0.22 μm PTFE membrane filter. The photoresist solution was spin-coated on a blank silicon wafer at a speed of 4000 r.p.m. for 90 s to form uniform films with the thickness of approximately 77 nm. The film was pre-baked at 100°C for 180 s. Electron beam exposure was carried out using an electron beam lithography system. After exposure, the film was baked at 120°C for 120 s. Development was applied in tetramethylammonium hydroxide (0.013 N, TMAH) at 20°C for 10 s and then followed by deionized water rinsing for 10 s.

## 2.3. Instruments

The thermal stability and glass transition temperature of BPA-6OH were characterized by TA Q50 thermogravimetric analysis (TGA) and TA Q100 differential scanning calorimetry (DSC), respectively. TGA and DSC were carried out at a heating rate of 10°C min$^{-1}$ under nitrogen atmosphere. Electron beam lithography was performed using a CABL-9000C electron beam lithography system with 50 keV acceleration voltage, 100 pA current, 300 μm field size and $60\,000 \times 60\,000$ dot @Vector scan (Crestec Corp.). Film thickness was measured by AST SE200BM spectroscopic ellipsometer. Scanning electron micrographs (SEMs) were recorded by Hitachi S8010 scanning electron microscopy. Atomic force microscopy (AFM) image was obtained by a Dimension Fast Scan instrument (Bruker) in tapping mode.

# 3. Results and discussion

## 3.1. The characterization of the BPA-6OH

Glass transition temperature (Tg) and thermostability are crucial factors in forming dense and stable thin films. High Tg and good thermostability are the essential requirements for the use in practical lithographic applications.

The TGA curve of BPA-6OH is shown in figure 2. For BPA-6OH, it began to lose weight at 154.6°C, which may be caused by residual solvent and moisture [35]. When the temperature reached 337.8°C, the first platform with the ratio of weight loss as 4.2% appeared, which demonstrated that BPA-6OH had high thermal stability.

The DSC thermogram of BPA-6OH is illustrated in figure 3. It clearly indicates that a step change around 120°C was observed, owing to the glass transition of the BPA-6OH. The thermal analysis mainly involves the baking process parameter. Different photoresists are matched with different baking process parameters, and sometimes different baking parameters can be applied to the same polymer-based photoresist. It is reported that the glass transition temperatures of some epoxide-based negative-tone molecular glass photoresists are generally lower than 100°C [36–38], while the glass

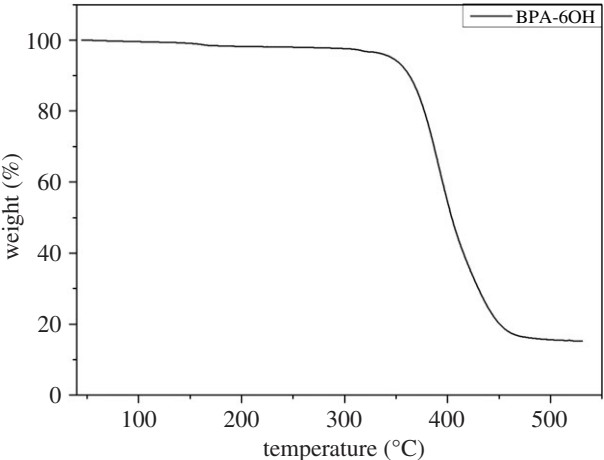

**Figure 2.** The TGA curve of BPA-6OH.

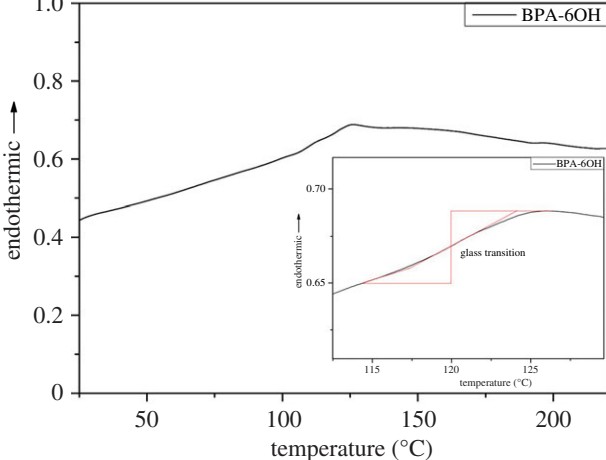

**Figure 3.** The DSC thermogram of BPA-6OH.

transition temperature of BPA-6OH negative photoresist is much higher. This high glass transition temperature can meet the requirement of the lithographic processes.

## 3.2. The possible reaction mechanisms of the BPA-6OH upon the electron beam exposure

Kozawa *et al.* proposed a radiation-induced reaction mechanism of a chemically amplified electron beam and extreme ultraviolet photoresists [39–41]. In this mechanism, the ion-molecular reaction between the polymer radical cation produced by the ionization of the polymer matrix after radiation and the polymer matrix was the primary pathway to acid generation. The electron excitation of the photoacid generator was the secondary pathway to acid generation. In the molecular glass photoresist, BPA-6OH radical cation and secondary electron were generated through electron beam irradiation. The BPA-6OH radical cation and the BPA-6OH formed BPA-6OH proton adduct, the photoacid generator reacted with the secondary electron to release anion, and the BPA-6OH proton adduct interacted with the anion to produce acid. The acid or proton reacted with the BPA-6OH and the TMMGU to form the BPA-6OH proton adduct and the TMMGU proton adduct, respectively. Then, the acid-catalysed reaction started, and the TMMGU underwent the cross-link reaction with the BPA-OH to give a negative-tone photoresist [42]. Upon the high-energy electron beam irradiation, a possible schematic diagram of the BPA-6OH negative-tone molecular glass photoresist can be drawn, as shown in figure 4.

## 3.3. Lithographic performance

The lithographic performance was affected by many factors, such as the compositions of the photoresist, baking temperature, spin coating, photoacid loading, exposure dose, development, and so on. For the

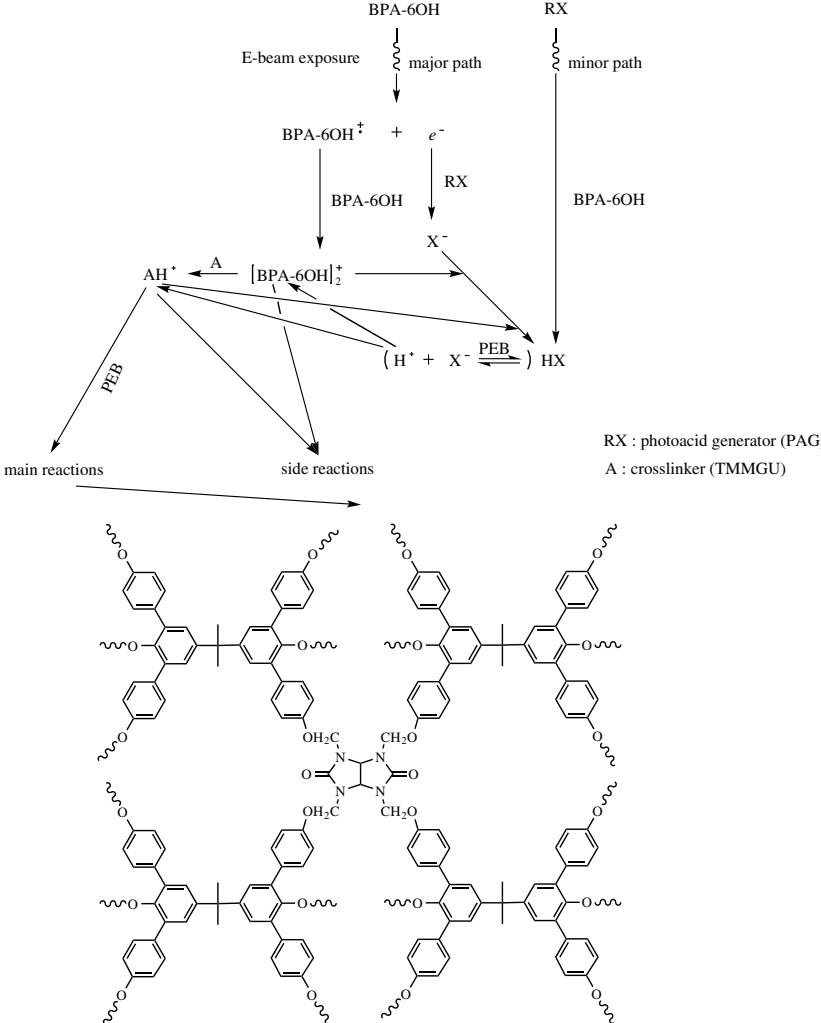

**Figure 4.** The possible reaction mechanisms of the negative-tone molecular glass BPA-6OH.

BPA-6OH photoresist, under the existing experimental conditions, exposure dose and post-exposure bake (PEB) temperature were the two most important controlling factors.

The post-exposure baking process may have a great impact on photoresist sensitivity and critical dimension control. In this process, the catalysation reaction of the photoacid generator and the cross-linking of the BPA-6OH with the TMMGU occurred, the solubility difference in the developer between exposure area and un-exposure area changed, which affected the quality of the patterns. We compared the performance under two PEB temperature conditions, 100°C and 120°C, as shown in figure 5. It can be seen that the pattern contrast was clear under these post-exposure bake temperature conditions, and the pattern quality of the higher PEB temperature treatment was better than that of the lower PEB temperature treatment at the same exposure dose, which indicated that it was more effective for the cross-link reaction at high PEB temperature.

In order to evaluate the optimal exposure dose range, we performed an exposure dose gradient experiment, fixing the acceleration voltage 50 keV and beam current 100 pA. An $8 \times 2$ exposure dose matrix was created with doses ranging from 32 to 92 $\mu$C cm$^{-2}$, and a series of $60 \times 300 \ \mu$m$^2$ rectangles was patterned with an increasing exposure dose. The different exposure doses were used to evaluate the effect of doses on the resolution and imaging performance at the PEB 120°C for 120 s. Figure 6 showed SEM images of line/space pattern of different CDs at different exposure doses. It can be seen that the BPA-6OH negative-tone photoresist was sensitive and 63.5 nm CD could be obtained at the exposure dose of 40 $\mu$C cm$^{-2}$. At an exposure dose of 72 $\mu$C cm$^{-2}$, the CD reached 85.3 nm, and had higher image contrast. We noticed that the patterns quality was poorer at low exposure doses than that at high exposure doses. It was visibly observed that the pattern distortion appeared at the exposure dose of 44 $\mu$C cm$^{-2}$, which may be caused by the insufficient cross-linking at the lower exposure dose. When the exposure dose exceeded 48 $\mu$C cm$^{-2}$, the patterns performances were

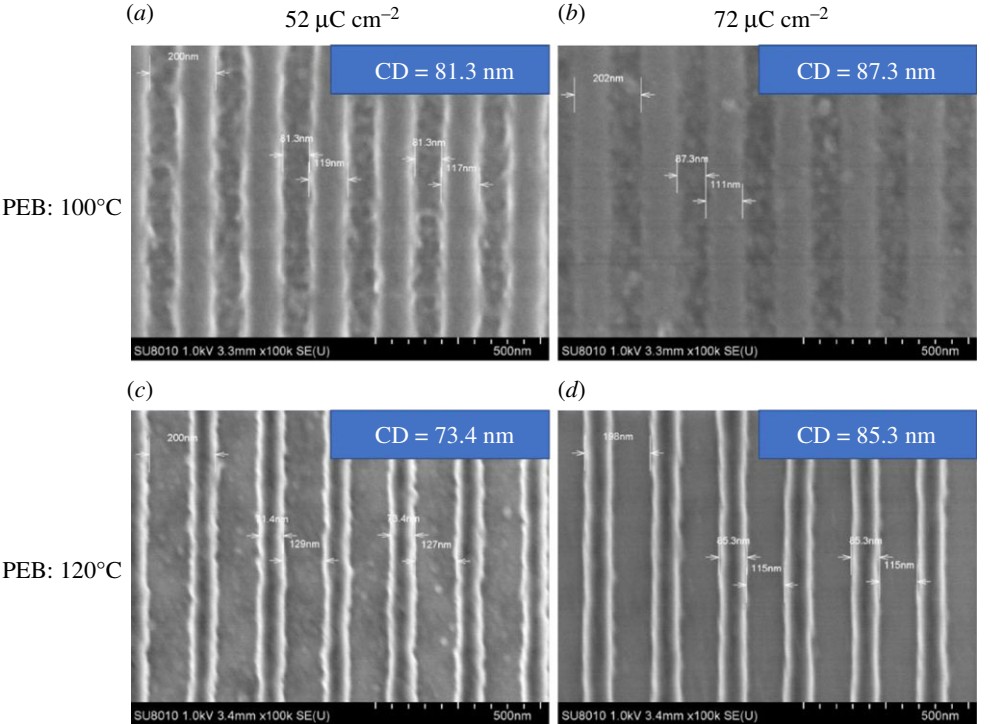

*(a)* 52 μC cm$^{-2}$  CD = 81.3 nm  *(b)* 72 μC cm$^{-2}$  CD = 87.3 nm

PEB: 100°C

*(c)*  CD = 73.4 nm  *(d)*  CD = 85.3 nm

PEB: 120°C

**Figure 5.** SEM images of lithographic performance evaluation of BPA-6OH resist at 100°C and 120°C PEB.

*(a)* 40 μC cm$^{-2}$  CD = 63.5 nm  *(b)* 44 μC cm$^{-2}$  CD = 71.4 nm  *(c)* 48 μC cm$^{-2}$

*(d)* 52 μC cm$^{-2}$  CD = 73.4 nm  *(e)* 56 μC cm$^{-2}$  CD = 81.3 nm  *(f)* 60 μC cm$^{-2}$

*(g)* 68 μC cm$^{-2}$  CD = 83.3 nm  *(h)* 72 μC cm$^{-2}$  CD = 85.3 nm

**Figure 6.** SEM images of line/space pattern of different CDs at different exposure doses.

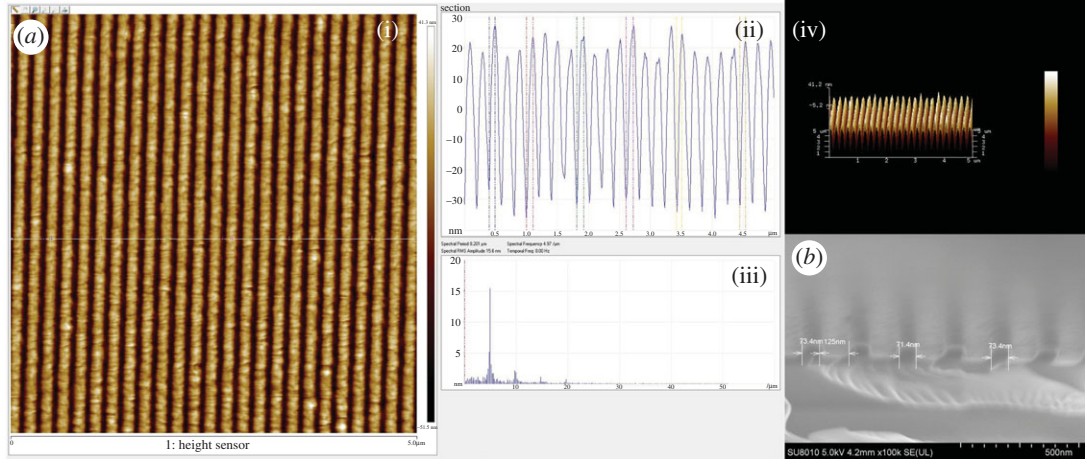

**Figure 7.** AFM and cross-section images of the negative-tone pattern at the exposure dose of 52 µC cm$^{-2}$ ((a) AFM image; (b) cross-section image).

improved. The resolution and image contrast were excellent as well as the CD reached 73.4 nm with the exposure dose of 52 µC cm$^{-2}$, and the corresponding AFM and cross-section images are shown in figure 7. It can be seen that the pattern had a good vertical profile and the vertical distance was about 56 nm (see electronic supplementary material, figures S3 and S4). When the exposure dose was 56 µC cm$^{-2}$, the resolution reached 81.3 nm. As the exposure dose increased to 68 µC cm$^{-2}$, 83.3 nm CD was received. If the exposure dose continued rising, it will lead to a great degree of cross-linking. This phenomenon proved that the effect of dose had a great impact on resolution and pattern performance. The exposure dose range 40–72 µC cm$^{-2}$ of the photoresist showed that BPA-6OH negative-tone photoresist had excellent sensitivity, high image contrast and good resolution.

## 4. Conclusion

We have shown that the BPA-6OH negative-tone photoresist was patterned under 50 keV electron beam and discussed the effect of exposure dose and PEB process on lithographic performance. The results demonstrated that the pattern quality at the higher PEB temperature was superior to the pattern quality at the lower PEB temperature due to the higher PEB temperature providing more activation energy for full cross-linking. It was found that it has both excellent resolution (73.4 nm CD), good sensitivity (52 µC cm$^{-2}$ at 50 keV electron beam) and a good vertical profile. It is possible to resolve smaller CDs, lower doses and higher image contrast by optimizing lithographic parameters and processes. Additional studies are underway to evaluate the optimal lithographic performances in practical application.

Data accessibility. The datasets supporting this article have been uploaded as part of the electronic supplementary material.
Authors' contributions. Y.W., L.C., J.Y. and X.G carried out the synthesis and electron beam lithography work. Y.W. prepared the manuscript, S.W. and G.Y. designed the study, coordinated the study and revised the manuscript. All authors gave final approval for publication and agree to be held accountable for the work performed therein.
Competing interests. The authors declare that they have no known competing financial interests or personal relationships that could have appeared to influence the work reported in this paper.
Funding. This work was supported by the National Natural Science Foundation of China (nos. 21873106, 21903085, 22073108, U20A20144 and 22090012) and the National Major Science and Technology Projects of China (no. 2018ZX02102).
Acknowledgements. The authors would like to acknowledge the kindly help from Dr. Shumin Yang and Dr. Jun Zhao in Shanghai Synchrotron Radiation Facility (SSRF).

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
