## [Peer Review File · Royal Society Open Science]

Review History

RSOS-202132.R0 (Original submission)

Review form: Reviewer 1

Is the manuscript scientifically sound in its present form?

Yes

Are the interpretations and conclusions justified by the results?

Yes

Is the language acceptable?

Yes

Do you have any ethical concerns with this paper?

No

Have you any concerns about statistical analyses in this paper?

No

Recommendation?

Accept with minor revision (please list in comments)

Comments to the Author(s)

The author reported a low-molecular-weight organic compound containing bisphenol A backbone as a negative photoresist. The author claims that the material has a high glass transition temperature, excellent thermal stability and good contrast. During electron beam exposure, the negative molecular glass photoresist can obtain a well-resolved line pattern of about 73.4 nm with a sensitivity of 52 $\mu\text{C}/\text{cm}^2$. Compared with traditional polymer photoresist, the molecular glass photoresist shows some advantages for advanced lithography.

The manuscript is well organized. Furthermore, the results are interesting, which should help to develop new photoresists with high performance. The art work and language of the manuscript needs to be improved. I recommend the publication of the manuscript after minor revisions.

Review form: Reviewer 2**Is the manuscript scientifically sound in its present form?**

Yes

Are the interpretations and conclusions justified by the results?

No

Is the language acceptable?

Yes

Do you have any ethical concerns with this paper?

No

Have you any concerns about statistical analyses in this paper?

No

Recommendation?

Accept with minor revision (please list in comments)

Comments to the Author(s)

The authors report a negative-tone molecular glass photoresist based on bis-phenol A backbone (BPA-6OH) for electron beam lithography. I find the manuscript interesting and well written for the readership. There are a few issues however that may need to be addressed:

- 1) It is mentioned in the article that this type of photoresist is a high-sensitivity chemically amplified photoresist, which is superior to non-chemically amplified photoresist in sensitivity. The author should provide some sensitivity data of typical non-chemically amplified photoresists, so that readers can be more specific understand the relevant differences.
- 2) The author should add some key data, such as the thermal analysis comparison of this type of photoresist with other photoresists, etc.
- 3) It is recommended to add some references on molecular glass photoresist to facilitate readers to better understand its development.

Decision letter (RSOS-202132.R0)

Dear Dr Wang:

Title: Negative-tone molecular glass photoresist for high resolution electron beam lithography
Manuscript ID: RSOS-202132

Thank you for submitting the above manuscript to Royal Society Open Science. On behalf of the Editors and the Royal Society of Chemistry, I am pleased to inform you that your manuscript will be accepted for publication in Royal Society Open Science subject to minor revision in accordance with the referee suggestions. Please find the reviewers' comments at the end of this email.

The reviewers and handling editors have recommended publication, but also suggest some minor revisions to your manuscript. Therefore, I invite you to respond to the comments and revise your manuscript.

Because the schedule for publication is very tight, it is a condition of publication that you submit the revised version of your manuscript before 29-Jan-2021. Please note that the revision deadline will expire at 00.00am on this date. If you do not think you will be able to meet this date please let me know immediately.

- 1) A text file of the manuscript (tex, txt, rtf, docx or doc), references, tables (including captions) and figure captions. Do not upload a PDF as your "Main Document".
- 2) A separate electronic file of each figure (EPS or print-quality PDF preferred (either format should be produced directly from original creation package), or original software format)
- 3) Included a 100 word media summary of your paper when requested at submission. Please ensure you have entered correct contact details (email, institution and telephone) in your user account
- 4) Included the raw data to support the claims made in your paper. You can either include your data as electronic supplementary material or upload to a repository and include the relevant doi within your manuscript
- 5) All supplementary materials accompanying an accepted article will be treated as in their final form. Note that the Royal Society will neither edit nor typeset supplementary material and it will

be hosted as provided. Please ensure that the supplementary material includes the paper details where possible (authors, article title, journal name).

Kind regards,
Dr Laura Smith
Publishing Editor, Journals

On behalf of the Subject Editor Professor Anthony Stace and the Associate Editor Professor Chaohua Cui.

RSC Associate Editor:
Comments to the Author:
(There are no comments.)

RSC Subject Editor:
Comments to the Author:
(There are no comments.)

Reviewer comments to Author:
Reviewer: 1

Comments to the Author(s)

The author reported a low-molecular-weight organic compound containing bisphenol A backbone as a negative photoresist. The author claims that the material has a high glass transition temperature, excellent thermal stability and good contrast. During electron beam exposure, the negative molecular glass photoresist can obtain a well-resolved line pattern of about 73.4 nm with a sensitivity of 52 uC/cm². Compared with traditional polymer photoresist, the molecular glass photoresist shows some advantages for advanced lithography.

The manuscript is well organized. Furthermore, the results are interesting, which should help to develop new photoresists with high performance. The art work and language of the manuscript needs to be improved. I recommend the publication of the manuscript after minor revisions.

Reviewer: 2

Comments to the Author(s)

The authors report a negative-tone molecular glass photoresist based on bis-phenol A backbone (BPA-6OH) for electron beam lithography. I find the manuscript interesting and well written for the readership. There are a few issues however that may need to be addressed:

- 1) It is mentioned in the article that this type of photoresist is a high-sensitivity chemically amplified photoresist, which is superior to non-chemically amplified photoresist in sensitivity. The author should provide some sensitivity data of typical non-chemically amplified photoresists, so that readers can be more specific understand the relevant differences.
- 2) The author should add some key data, such as the thermal analysis comparison of this type of photoresist with other photoresists, etc.
- 3) It is recommended to add some references on molecular glass photoresist to facilitate readers to better understand its development.

Author's Response to Decision Letter for (RSOS-202132.R0)

See Appendix A.

Decision letter (RSOS-202132.R1)

Dear Dr Wang:

Title: Negative-tone molecular glass photoresist for high resolution electron beam lithography
Manuscript ID: RSOS-202132.R1

It is a pleasure to accept your manuscript in its current form for publication in Royal Society Open Science. The chemistry content of Royal Society Open Science is published in collaboration with the Royal Society of Chemistry.

On behalf of the Subject Editor Professor Anthony Stace and the Associate Editor Professor Chaohua Cui.

RSC Associate Editor
Comments to the Author:
(There are no comments.)

Reviewer(s)' Comments to Author:

Appendix A

Reply to Reviewer comments to Author:

Reviewer: 1

Comments to the Author(s)

The author reported a low-molecular-weight organic compound containing bisphenol A backbone as a negative photoresist. The author claims that the material has a high glass transition temperature, excellent thermal stability and good contrast. During electron beam exposure, the negative molecular glass photoresist can obtain a well-resolved line pattern of about 73.4 nm with a sensitivity of 52 $\mu\text{C}/\text{cm}^2$. Compared with traditional polymer photoresist, the molecular glass photoresist shows some advantages for advanced lithography.

The manuscript is well organized. Furthermore, the results are interesting, which should help to develop new photoresists with high performance. The art work and language of the manuscript needs to be improved. I recommend the publication of the manuscript after minor revisions.

Reply: Thank you very much for your suggestions. We have carefully modified the manuscript in grammar, sentences structure, etc.

Reviewer: 2

Comments to the Author(s)

The authors report a negative-tone molecular glass photoresist based on bis-phenol A backbone (BPA-6OH) for electron beam lithography. I find the manuscript interesting and well written for the readership. There are a few issues however that may need to be addressed:

1) It is mentioned in the article that this type of photoresist is a high-sensitivity chemically amplified photoresist, which is superior to non-chemically amplified photoresist in sensitivity. The author should provide some sensitivity data of typical non-chemically amplified photoresists, so that readers can be more specific understand the relevant differences.

Reply: In the chemically amplified resist (CAR) system, the catalyst, such as the photoacid, is produced by the photoacid generator undergoing the photochemical reactions after irradiation, which can interact with the surrounding matrix to trigger a cascade of chain reactions. Consequently, the multiple resist exposure events can be caused by a single irradiation, thereby increasing the photosensitivity. Compared with Non-CAR, the number of photons, or dose, required to achieve equivalent solubility

change in CAR is greatly reduced. For polymer-based resist, such as polymethylmethacrylate (PMMA), the researcher found that the sensitivity of polymer-based resist primarily depends on the nature of the polymer, molecular weight and molecular weight distribution, and it is also influenced by the lithographic processes. PMMA is well-known as the industry standard high-resolution resist for e-beam lithography. The sensitivity of positive-tone PMMA was reported to be about 350 $\mu\text{C}/\text{cm}^2$ using 50 keV acceleration voltage (reference 31).

2) The author should add some key data, such as the thermal analysis comparison of this type of photoresist with other photoresists, etc.

Reply: Thermal analysis mainly involves the baking process parameter. Different photoresists are matched with different baking process parameters, and sometimes different baking parameters can be applied to the same polymer-based photoresist. It is reported that the glass transition temperatures of some epoxide-based negative-tone molecular glass photoresists are generally lower than 100 °C (reference 36, 37, 38), while the glass transition temperature of BPA-6OH negative photoresist is much higher. This high glass transition temperature can meet the requirement of the lithographic processes.

3) It is recommended to add some references on molecular glass photoresist to facilitate readers to better understand its development.

Reply: Thank you for your valuable suggestions. We have added the reference 21, 22, which mainly introduce molecular glass photoresist materials. In fact, we have cited some literatures about the molecular glass photoresist from Professor CK Ober' group of Cornell University, e.g., references 12, 17, 18, 19, 20. These articles describe some properties of molecular glass material and the design principles of molecular glass photoresist. We believe that the outstanding work of Professor CK Ober' group in the field of lithography will be highly beneficial to those who are interested in lithographic materials. Of course, other author's excellent works, such as DP Sanders in reference 11, are also very enlightening and representative.